# Higher Fruit and Vegetable Intake Is Associated with Participation in the Double Up Food Bucks (DUFB) Program

**DOI:** 10.3390/nu13082607

**Published:** 2021-07-29

**Authors:** Abiodun T. Atoloye, Mateja R. Savoie-Roskos, Carrie M. Durward

**Affiliations:** 1UCONN Rudd Center for Food Policy and Obesity, One Constitution Plaza, Suite 600, Hartford, CT 06103, USA; abiodun.atoloye@gmail.com; 2Department of Nutrition, Dietetics and Food Sciences, Utah State University, 8700 Old Main Hill, Logan, UT 84322, USA; mateja.savoie@usu.edu

**Keywords:** farmers’ market incentives, fruit and vegetables, fruit and vegetable incentives, Supplemental Nutrition Assistance Program (SNAP), fruit and vegetable consumption, low-income populations

## Abstract

Incentivizing fruit and vegetable (F&V) purchases may help address barriers to healthy eating among populations with low income. In a repeated measures natural experiment study, we examined whether participation in the Double Up Food Bucks (DUFB) program increased F&V consumption among Supplemental Nutrition Assistance Program (SNAP) recipients. Two hundred and twelve participants recruited at baseline through telephone calls were informed about the availability of DUFB at their local farmers’ market (FM). F&V consumption frequency and DUFB use were obtained at baseline, mid FM, and end of FM season approximately 5 months later. Participants (*N* = 212) were primarily white (76.4%) women (77.3%) with an average age of 43.5 years. Only 34 participants opted to use the DUFB program. A linear mixed model showed a significant main effect of DUFB use (*p* = 0.001) and of time (*p* = 0.002), with a decrease in F&V intake over time. Compared to non-users, DUFB users had a significantly higher F&V consumption at baseline and midpoint (*p* = 0.02 and *p* = 0.02, respectively). F&V consumption was associated with participation in the DUFB program and higher F&V consumption frequency was observed prior to program use among program participants. Future interventions that specifically target SNAP recipients with low F&V intake to use the DUFB program are needed.

## 1. Introduction

Incentivizing purchases at farmers’ markets, supermarkets, and other food stores is a governmental, non-profit, and corporate effort to improve the dietary intake of the low-income population [1,2]. Double Up Food Bucks (DUFB) is one such program established to encourage fruit and vegetable (F&V) intake among Supplemental Nutrition Assistance Program (SNAP) recipients while also supporting local farmers [3,4]. Programs that give price incentives, coupons, or vouchers at point of purchase offer opportunities to increase access [5], purchases [6,7], and F&V consumption [8,9,10] among low-income populations. However, previous literature has identified several barriers to SNAP recipients’ use of such programs including lack of program awareness, inconvenient location/operation hours, and lack of transportation [11,12,13].

The available evidence on the effect of farmers’ market (FM) incentive programs on F&V consumption has been mixed with the majority of studies suggesting a positive association between F&V consumption and incentive programs. A cross-sectional study found that DUFB users are more likely to try a new fruit and have a higher F&V intake than non-users (OR = 1.8, *p* = 0.006 and OR = 2.4, *p* = 0.001, respectively) [8]. A significant increase in the intake of some vegetables by DUFB users (*p* = 0.001) was found in a pretest–posttest design [10]. Another pretest–posttest study found a 0.4 times-per-day increase in F&V intake among participants in a DUFB program (*p* = 0.002) [14]. A different study using a pretest–posttest design found that incentive program participants with lower baseline F&V consumption were more likely to increase their F&V consumption after participation in an incentive program [15]. A repeated measure, clinic-based intervention study of 177 diabetic patients receiving SNAP benefits found a positive association between FM incentive use and increased F&V consumption (*p* = 0.002) [16]. A quasi-experimental study assessed the effect of an FM incentive program, by looking at neighborhoods with and without FM incentives in the same city [11]. The authors did not find a significant association between F&V consumption and the presence of incentive programs. However, a cross-sectional analysis of the same intervention found that Health Bucks users were more likely to self-report an increase in the F&V intake compared to the previous year [11]. A cross-sectional study among 288 participants of the Produce Plus Program did not find a significant difference in F&V consumption by frequency of incentive use [17]. Another cross-sectional study found an impact of the Double Up FM incentive program on the self-reported number of F&V but not the frequency of F&V consumption [18].

The available evidence tends to indicate that these farmers’ market incentive programs may impact consumption, however, many of these studies lack true control groups or comparison of outcomes with non-users. Overall, the evidence is mixed and lacks rigorous study designs. Based on these limitations, the available evidence is not sufficient to establish strong conclusions about the impact of FM incentive participation on increased F&V consumption. Studies that would provide stronger conclusions, such as those that include control groups with comparable baseline characteristics, are needed [19]. This study examined whether DUFB program participation was associated with increased F&V consumption among SNAP recipients using a natural experiment design, or group assignment without the influence of the researcher.

## 2. Materials and Methods

### 2.1. Utah DUFB Program (Intervention)

In 2016, the Utah DUFB program was available at 21 outlets (18 farmers’ markets, 2 farm stands, and 1 mobile market) at 28 locations in 11 counties across the state. Nearly 90% of the SNAP recipients in the state of Utah resided in these 11 counties, and about two thirds of SNAP recipients lived within 5 miles of a DUFB location in 2016. In 2016, the program provided a dollar-per-dollar match of SNAP spent at participating locations, up to USD 10 per FM visit. To receive DUFB at most locations, participants: (1) would go to a market information booth at a participating FM, (2) tell the booth attendant how much they wanted to spend in SNAP funds at the FM and scan their Electronic Benefit Transfer card for a purchase of that amount, (3) receive tokens valid for any SNAP eligible product at the market in the amount specified in step 2 and (4) automatically receive the same amount (up to USD 10) in DUFB tokens which were only valid for F&V purchases at the market. In addition to other marketing strategies, more than 52,000 SNAP recipient households that lived within 5 miles of a DUFB location were mailed a flyer describing the program in 2016. Because previous work indicated that a major barrier to incentive program participation was a lack of awareness, we also informed study participants about the program and nearby locations during our baseline data collection call. This informational strategy can be thought of as a secondary intervention, while the primary intervention of interest was participation in the DUFB program. 

### 2.2. Recruitment and Data Collection

We recruited study participants from SNAP recipients who resided in Utah. An *a priori* power calculation indicated that a sample size of about 410 would be needed to evaluate change in F&V consumption. This calculation assumed a between groups mean difference of 0.5 and standard deviation of 0.8, with alpha set to 0.05 (G*Power version 3.1, Heinrich Heine Universitat Dusseldorf, Dusseldorf, Germany, 2014). This resulted in a per-group sample size of 41 needed for 0.8 power. To determine initial sample size needed, we estimated 40% retention from baseline to endpoint based on previous work [10,14], and made an educated guess that ¼ of the participants recruited would choose to use the DUFB program, meaning that we would need to recruit 410 participants. Contact information for 4000 SNAP recipients was obtained from the Utah Department of Workforce Services, based on previous unpublished recruitment efforts of SNAP recipients, which resulted in about a 17% response rate. Letters were sent to all the 4000 recipients and phone calls were made to 1265 of those who received letters during the recruitment period. Of those who were called and mailed letters, 434 were reached. Of those reached, 212 SNAP recipients participated in the study, 175 declined, and 47 were ineligible. 

Eligible participants were ≥18 years of age, current recipients of SNAP benefits, willing to participate in the study, and residents in the study area. Data were collected from 161 of these individuals at the midpoint (76% of the initial sample), and 123 participated at the endpoint (58% of the initial sample). Of these, 111 participants had complete data. All data were self-reported and obtained by an interviewer over the phone. 

### 2.3. Measures

At baseline, which occurred before or at the beginning of the FM season (9 June–10 August 2016) we asked participants if they were aware of the DUFB program. If they said no, they were given a brief description of the program and its benefits as well as information about the closest participating FM. Other information obtained at the baseline included the willingness to use the program in the future, use in previous season, and frequency of use, if applicable. For the midpoint and endpoint surveys (3 August–19 September (mid-season) and 21 September–3 November (end of season), 2016, respectively), participants were asked whether they had shopped at a FM since the last contact, how frequently they shopped at the FM, and if they used DUFB at the FM (with options of “no,” “sometimes but not all of the times,” and “yes, every time”).

F&V consumption data were obtained at the baseline, midpoint, and endpoint using the F&V consumption measures from the Behavioral Risk Factor Surveillance System (BRFSS) [20], a 6-item dietary screening tool that assesses the frequency of consuming 100% fruit juice, fruit, beans (legumes), dark green vegetables, orange vegetables, and other vegetables over the past month [21]. The tool has been shown to have moderate validity compared to multiple 24-h recalls or diet records [22]. Information on participants’ demographics was also obtained. Participants were given monetary incentives for participating in the survey including USD 10 for baseline, USD 20 for midpoint, and USD 30 for endpoint. 

For analysis purposes, participants were grouped into two categories: DUFB users and non-users. Participants received a “1” for DUFB use if the participant reported using DUFB at either midpoint, endpoint, or both times, otherwise “0” (non-users) was assigned. DUFB use was the independent variable, while F&V consumption was the dependent (time-varying) variable. F&V consumption responses from the F&V screener were given in frequencies of consumption by either day, week, or month; these responses were converted to daily frequencies [21].

### 2.4. Analysis

Demographic characteristics were compared between: (1) those who completed the survey at midpoint and endpoint (*N* = 161 and *N* = 123, respectively) and those who did not complete the survey at either of those two time points, (2) those with complete data (*N* = 111) and those without complete data, and (3) DUFB users and non-users (the experimental groups). Chi-square tests were used for categorical variables and two-sided independent *t*-tests were used for continuous variables. Data are presented as mean ± standard deviation (SD) for continuous variables and *N* (%) for categorical variables.

To examine the main effect of DUFB participation on F&V consumption and the difference in F&V consumption among DUFB users and non-users after baseline, we generated linear mixed-effect models. First, we ran an empty model (using only the F&V consumption variable) to estimate the intra-cluster correlation coefficient (ICC) (a reliability rating for clustering effect in the outcome variable). We found an ICC of 0.78, which reflected that the variance in F&V intake could be explained by the multiple observations collected from the same individual. Time and DUFB use were added to the model as fixed effects so that the change in F&V consumption between DUFB users vs. non-users and across each time point could be assessed by the coefficients or estimated using the least square means. The missing data points were assumed missing completely at random, and no imputation of missing data was carried out. The model allowed data from participants with some missing data to be included, in addition to those with complete data (496/636 observations used). The Huynh-Feldt variance structure was assumed and estimation was by the restricted maximum likelihood. In an adjusted model, we controlled for age, gender, race/ethnicity, household size (number of adults and children in the household), income, and education.

To account for possible confounding effect on the outcome from individuals who had already used DUFB at baseline, a sensitivity analysis was run using the above linear mixed-effect model but without data from participants who were baseline users of DUFB program (*N* = 15). For this analysis, 459/591 observations were used.

The second sensitivity analysis explored the relationship between frequency of use of DUFB and F&V consumption. We combined participant’s responses to two questions to create a variable for DUFB use frequency: (i) “How many times have you shopped at a farmers’ market over the past month?” “0”, “1”, “2–3”, “4–5”, “6–7”, “8–9”, or “>9” times and (ii) “When you shopped at the farmers’ market did you use your EBT card and get matching DUFB tokens to purchase extra produce?” “no”, “sometimes but not all of the times”, and “yes, every time”. The number of times participants shopped at FM was recoded as “0” or the upper limit (e.g., “2–3” as “3” etc.). How often DUFB was received when shopping at the FM was recoded: “no” as “0”, “sometimes but not all of the times” as “0.5”, and “yes, every time” as “1”. Then, both variables were combined by multiplication for each time point (midpoint and endpoint) and both were summed to give a proxy variable for the frequency of use, with resulting values ranging between 0.5 and 10. Frequent users were classified as those with a frequency score between 5–10 and less frequent users had a frequency score of less than 5. We examined the difference in F&V intake among just DUFB users; *N* = 34 (frequent users, *N* = 10 and less frequent users, *N* = 24) in another linear mixed-effect model (92/102 observations). All the analyses were performed using SAS v. 9.4. (SAS Institute, Cary, NC, USA, 2013), with the significance level set at 0.05. 

## 3. Results

### 3.1. Participant Demographic Information

At baseline, we recruited 212 participants who were primarily white (76.4%, *N* = 162), women (77.4%, *N* = 164), with an average age of 43.53 years (±15.63 SD). Table 1 reports the demographic characteristics of participants who completed the survey at the midpoint and endpoint as well as the group of participants with complete data. In addition, Table 1 shows differences between these participants and those who did not complete the survey at midpoint, endpoint, or have complete data, respectively. Among demographic characteristics tested, there were significant differences in age for participants who completed the survey at the endpoint versus those who did not (*p* = 0.01). Those who had complete data (*N* = 111, 52% retention rate) also had higher mean age (by 5.4 years) compared to those lost to attrition (*p* = 0.01). Education level was significantly higher for participants who completed the midpoint survey versus those who did not and for participants with complete data versus those lost to attrition (*p* = 0.05 and *p* = 0.01 respectively). However, as shown in Table 2, there were no significant demographic differences between DUFB users and non-users. 

### 3.2. Farmers’ Market Incentives Usage

Among the total 212 participants, 40 (18.9%) participants were already aware of the DUFB program at baseline with 15 (7.1%) reporting that they were DUFB users at baseline (before the FM season). A total of 34 participants used DUFB after we informed them of the program (16.0%). Among the 15 participants who were previous DUFB users at baseline, only six continued to use DUFB. The other 28 were new DUFB users in the 2016 season.

### 3.3. Fruit and Vegetable Consumption

Table 3 and Figure 1 present the results from the linear mixed model using the coefficient estimate and least square means, respectively, by DUFB participation/group for each time point. The main effects of time and DUFB use were statistically significant (*p* = 0.002, *p* = 0.001, respectively). Further, the interaction effect of both time and DUFB use was not significant (*p* = 0.43), indicating that the relationship between DUFB participation on F&V consumption was consistent over time. DUFB users had significantly higher F&V consumption frequency than non-users at both baseline and midpoint (*p* = 0.02, *p* = 0.02, respectively.) However, the difference in their F&V consumption at the end of the FM season was not significant. Additionally, there was a significant within group difference in the F&V consumption reported at baseline and endpoint for both DUFB users and non-users (*p* = 0.01, *p* = 0.01, respectively) while the difference between baseline and midpoint intake was only significant for non-users (*p* = 0.01). In the adjusted model, the main effects of time and DUFB use remained statistically significant (*p* = 0.02, *p* = 0.02, respectively) and the interaction effect of both time and DUFB use remained not significant (*p* = 0.53). This indicates that the impact of DUFB participation on F&V consumption was consistent over time irrespective of the demographic characteristics of the study participants.

The first sensitivity analysis without data from participants who were previous users of the DUFB program indicated that the main effects of time and DUFB use were statistically significant (*p* = 0.005, *p* < 0.001, respectively). Further, the interaction effect of both time and DUFB use was not significant (*p* = 0.54), indicating that the impact of DUFB participation on F&V consumption was consistent over time. These findings are similar to the findings when the full dataset was used. However, DUFB users had significantly higher F&V consumption frequency than non-users at all the time points in this analysis (*p* = 0.001, *p* = 0.001, and *p* = 0.03, respectively). 

The second sensitivity analysis of the difference in the F&V consumption between frequent DUFB users (*N*=10) and less frequent users (*N* = 24) did not find a significant difference in F&V consumption by time and the frequency of use (*p* = 0.12, *p* = 0.14).

## 4. Discussion

Although previous longitudinal studies have examined associations between FM incentives and F&V consumption among SNAP recipients [16,23], this study is the first to examine the association between FM incentive use and F&V consumption among SNAP recipients using a natural experiment longitudinal study. Our main finding was that DUFB participation was associated with F&V consumption, with higher F&V consumption among DUFB users compared to non-users. In addition, the difference between the DUFB users and non-users F&V intake was consistent at all time points.

This study used an informational strategy to address the commonly reported barrier of lack of awareness of FM incentive programs. With a change from 7.1% to 16.0% in DUFB use among the study population, the information strategy used appears to be helpful in getting SNAP recipients to participate in the FM incentive program, similar to a previous study [16]. The change in incentive program use is comparable in magnitude to a multi-state incentive evaluation, which found that 14.6% of their general farmers’ market sample used incentives during the intervention period, up from 11.5% who had previously received incentives before the study [24]. However, the results presented here indicate a simple information strategy may not be sufficient to motivate usage among SNAP recipients who do not already consume F&V. These individuals may not value the FM incentive program as highly, or experience other barriers such as limited transportation or inconvenient FM operation hours, among others [13,23,24]. Among participants who had not previously used the program, 15% reported use after our brief informational intervention, which is much lower than the previously mentioned clinical intervention, which resulted in 61% of participants using the program [16]. This difference may be because our intervention was delivered orally over the phone while the intervention in the previous study included in-person oral explanations, written materials, a map, and also included a USD 10 voucher [16]. Using an informational strategy in combination with other strategies (for example, a combination of cooking demonstration and marketing strategy) as reported in a previous study [18] might be useful to reduce barriers or increase perceived value among SNAP recipients. 

Similar to previous studies that reported positive associations between FM incentive program use and F&V consumption [8,10,14,18,23], we observed higher F&V consumption among DUFB users compared to non-users at baseline and midpoint. However, we did not see an increase in F&V consumption after use of the DUFB program. In line with our findings, one longitudinal study did not find a significant change in F&V consumption due to incentive use over the course of the FM season [18]. Our findings contrast with previous longitudinal studies that showed an increase in F&V consumption after program participation [10,14,15,16]. However, the mixed methods study we previously discussed had different results for each arm of the study: a cross-sectional intercept survey and a random-digit phone call survey [11]. The intercept survey targeted shoppers at both participating and non-participating FMs and found a self-reported increase in F&V consumption due to program participation. The other arm included the combination of primary and secondary multi-year data obtained via a phone call survey among residents of neighborhoods with and without the incentive program. Using a difference-in-difference approach, this longitudinal study found no detectable difference in F&V consumption among residents in the participating neighborhoods compared to non-participating neighborhoods [11]. The differences in the findings across these studies may be due to the different measurement tools used, population differences, sample size, or the bias inherent in the study designs used leading to differences in participants’ responses.

In contrast to one of the most rigorously designed studies of F&V incentives, the USDA Healthy Incentive Pilot (HIP) [25], the current study did not observe an increase in consumption after participation. Instead, there was a significant between-group difference in the mean number of times F&V were consumed in a day (1.13 ± 0.35 SD (*p* = 0.001), about 38% higher) for DUFB users vs. non-users at baseline. The HIP study found a 0.24 cup-equivalent-per-day (26% higher than non-participants, *p* < 0.001) increase in F&V consumption due to participation in a 30% rebate program for targeted F&V purchase in grocery stores [25]. (Although the HIP incentives were technically available at FM, <0.5% of study purchases were made at FM [25].) The difference in outcomes between studies might be due to the study setting (grocery store vs. FM), the form and amount of incentives, different measurements of F&V consumption (Automated Multiple-Pass Method 24-h dietary recall cup equivalent per day vs. BRFSS number of times per day), or different experimental designs (randomized vs. self-selection to treatment). Both studies provide useful information on the impact of incentive programs among SNAP recipients and indicate that program impacts may be different depending on study setting and form/amount of incentive.

Although frequent DUFB users had higher F&V consumption frequency by 0.74 time per day on average than less frequent users during the study period, their F&V intake did not differ significantly in the second sensitivity analysis. However, this result should be interpreted with caution because of the extremely small sample size. A previous study found that frequent FM shoppers received a larger quantity of DUFB tokens over the market season, which theoretically should result in larger F&V purchase and consumption [26]. 

Although this study does not provide evidence to help explain what might be responsible for the observed significant difference in F&V consumption between DUFB users and non-users at baseline (before DUFB awareness was created), previous literature indicates that F&V consumption and FM shopping are strongly associated. A cross-sectional study associated higher intake (by about 1.3 servings) of F&V with frequent FM shopping (at least 2–3 times per month) compared to less frequent FM visits (at most once a month, *p* = 0.03) [27]. Another study found a significant, positive association between F&V intake and FM shopping among SNAP recipients (*p* = 0.001). Further, another study found higher odds of improved F&V consumption among diabetic patients with a higher frequency of FM visits compared to those with less frequent visits (OR: 2.1, 95% CI 1.1, 4.0) [23]. The high baseline F&V intake of about 4.62 times per day among DUFB users may suggest that those who value F&V may have more interest in overcoming barriers to F&V use.

This study suggests that the DUFB program may contribute to increased access to F&V among people who already consume or value F&V. If the program is able to reach SNAP recipients who are low F&V consumers, it may lead to an improved F&V consumption as some previous studies have suggested [8,10,14]. More resources and effort may be needed to reach SNAP participants with low baseline F&V consumption and help them overcome barriers to participation.

Surprisingly, there was an unexpected decrease in F&V consumption frequency over time in both groups, in contrast to the hypothesized increase in DUFB users. This trend may be due to response shift bias where participants may have overstated their baseline F&V consumption responses and realigned their responses due to the study/intervention. The literature has suggested using a retrospective post-then-pretest method of data collection to reduce this effect [28]. Another potential cause may be seasonality of F&V consumption [29]. 

Limitations include the difference in the number of participants that used DUFB vs. non-users and the fact that factors that could influence outcomes (e.g., income, distance, shopping at FM, using SNAP benefit at FM) were not accounted for in the statistical model used. While the two study groups (DUFB users and non-users) were demographically similar, they were different in F&V consumption at baseline and other unmeasured criteria that caused them to not choose to use the program. Although the targeted sample size was not reached, further recruitment calls were not possible because of staff time limitations, and the fact that the FM season was starting. Further years of data collection were not possible because the recipient contact information was no longer available to the researchers. This may have resulted in insufficient power. About 22% of the data for the outcome variable were missing, which is not unusual for a longitudinal study [30]. However, these data were handled by an appropriate data analysis method (mixed model). The education status and age differed by survey completion status and this could limit the generalizability of the findings [31]. This study had a very low overall response rate (4000 recruitment letters resulted in 212 participants), which probably resulted in selection bias. Further, we are not able to compare our sample to non-respondents because demographics were not provided. However, demographic characteristics of the total population of SNAP recipients can be used as an indicator. The sample with complete data was comparable to the general Utah SNAP population on race/ethnicity (87.3% vs. 89.3% non-Hispanic whites, respectively) [32]. Despite the retention strategies (high monetary incentives, low participant burden, and frequent contact with study staff) used in this study, there was a low retention rate (52%). Low retention rate is common in studies among populations with low income [33]. Although we asked participants for multiple forms of contact information, a majority only provided a single phone number and mailing address. Multiple contacts from all participants might increase the chance of reaching some of those lost during follow up [16]. 

However, despite limitations, this study has value because it follows participant through time in a natural experiment study. Longitudinal studies of DUFB incentive type programs are not typical, instead most are cross-sectional. Other longitudinal studies have examined incentives at grocery stores [25] rather than FM or looked only at F&V purchase rather than consumption [24]. Further, previous studies used an intercept survey method at FMs, which is also prone to selection bias (you can only recruit current farmers’ market shoppers), limits the generalizability of the results, and may be inconvenient for respondents [8,10,14]. The current study used a data collection method that reduces the likelihood of selection bias and the inconvenience inherent in the intercept survey method. 

Another strength of this study lies in the ecological validity of the setting used. Two previous longitudinal studies recruited diabetic patients in a clinic-based intervention as participants; this choice of recruitment method and intervention setting may limit the generalizability of the findings and may be associated with social desirability bias [16,23]. The recruitment method and settings used in this study represent the kind of setting in which the decision about whether to use DUFB is up to the SNAP participant. It is worth noting that obtaining a more ecologically valid study population is likely to yield more reliable findings on the efficacy of dollars going to FM incentives. 

## 5. Conclusions

These results do not support the hypothesis that DUFB participation increases F&V consumption. Rather, in our sample, SNAP recipients who chose to use the program already had higher F&V consumption frequency before program use. The decrease in F&V consumption frequency over time in both groups may be due to response shift bias. Future studies may consider using the retrospective post-then-pretest method to reduce the effect of this bias. In addition, future analyses should control for factors that could influence F&V consumption among this population. Some caution should be used when interpreting these results because of the low retention rate and small sample size in the treatment group. The findings indicate that, while DUFB may contribute to increased access to F&V among high F&V consumers, more work may be needed to reach SNAP participants with low baseline F&V consumption. These results raise the possibility that interventions to increase value and decrease barriers for F&V among SNAP recipients with low F&V consumption are necessary to increase participation, in addition to simply raising awareness of the incentive programs. Further study to understand the differences between SNAP participants who choose to use FM incentive programs like DUFB and those who do not would be valuable to help target future interventions. 

## Figures and Tables

**Figure 1 nutrients-13-02607-f001:**
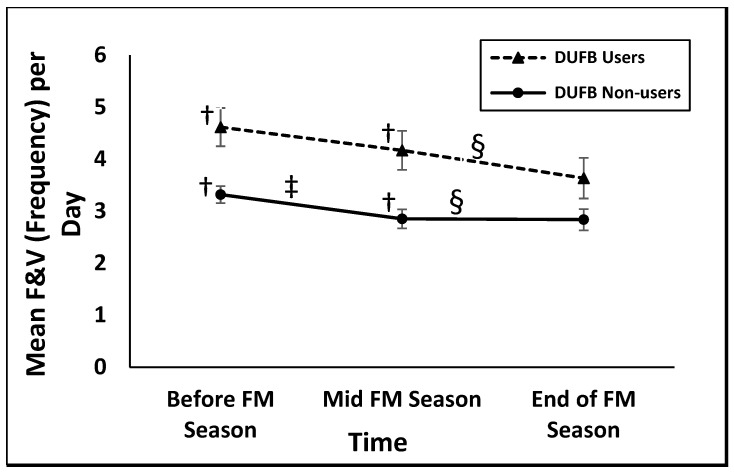
Fruit and Vegetable (F&V) Consumption among Double Up Food Bucks (DUFB) Users and Non-users in 2016 Farmers’ Market Season (*n* = 212) (using 496 observations out of 636). F&V, Fruits and Vegetables; DUFB, Double Up Food Bucks; FM, Farmers’ Market. † indicates a significant difference in F&V consumed by DUFB users and non-users at baseline and mid FM season (*p* < 0.01). ‡ indicates a significant difference in F&V consumption from baseline to mid FM season (*p* < 0.01). § indicates a significant difference in F&V consumption from baseline to the end of FM season (*p* < 0.05).

**Table 1 nutrients-13-02607-t001:** Participant Demographics by Survey Completion.

	Midpoint (*N =* 161)		Endpoint (*N* = 123)		Complete Data (*N* = 111)	
	Mean (SD)	*p*-Value	Mean (SD)	*p*-Value	Mean (SD)	*p*-Value
Variables						
Age	43.98 (15.9)	0.46	46.13 (16.28)	**0.01**	46.12 (6.34)	**0.01**
Household size *	3.2 (2.1)	0.23	2.9 (2.1)	**0.01**	2.9 (2.2)	**<0.01**
	***n* (%)**		***n* (%)**		***n* (%)**	
Gender		0.83		0.70		0.71
Female	124 (77.0)		94 (76.4)		87 (78.4)	
Male	37 (22.9)		29 (23.6)		24 (21.6)	
Race/ethnicity		0.12		0.06		**0.02**
White	132 (82.0.)		101 (82.1)		94 (84.7)	
Black	4 (2.5)		1 (0.81)		1 (0.9)	
Hispanic	16 (9.4)		14 (11.4)		9 (8.1)	
Non-Hispanic other	9 (5.6)		7 (5.7)		7(6.3)	
Education		**0.05 ***		0.26		**0.01**
≤Grade 11	11 (9.1)		14 (11.5)		10 (8.2)	
>Grade 11	97 (79.5)		105 (86.1)		97 (79.5)	
Household income		0.61		0.43		0.71
≤USD 20,000	89 (83.9)		97 (82.9)		88 (83.8)	
>USD 20,000	17 (14.2)		20 (17.1)		17 (16.2)	

* Household size = number if adults + children in the household; *p*-values indicate differences between the participants in the column and those who did not complete the survey at midpoint, endpoint, or have complete data, respectively. Bolded values highlight *p*-values ≤ 0.05.

**Table 2 nutrients-13-02607-t002:** Participant Demographics by Double Up Food Bucks (DUFB) Participation (*N* = 212).

	DUFB Users (*n =* 34)	DUFB Non-Users (*n =* 178)	
Variables	Mean (SD)	Mean (SD)	*p*-Value
Age	43.3 (14.84)	43.6 (15.8)	0.92
Household size *	3.1 (1.6)	3.3 (2.2)	0.59
	***n* (%)**	***n* (%)**	
Gender			0.75
Female	27 (79.4)	137 (77.0)	
Male	7 (16.5)	41 (23.0)	
Race/ethnicity			0.67
White	27 (79.4)	135 (75.8)	
Black	2 (5.9)	5 (2.4)	
Hispanic	3 (8.8)	23 (12.9)	
Non-Hispanic other	2 (5.9)	15 (8.4)	
Education (*n =* 122)			0.98
≤Grade 11	3 (11.1)	12 (12.6)	
>Grade 11	24 (88.9)	83 (87.4)	
Household income (*n =* 120)			0.14
≤USD 20,000	20 (74.1)	80 (86.0)	
>USD 20,000	7 (25.9)	13 (14.0)	

DUFB, Double Up Food Bucks; SD, Standard Deviation; * Household size = number if adults + children in the household.

**Table 3 nutrients-13-02607-t003:** Association of Double Up Food Bucks Program and Fruits and Vegetable Intake Among Program Participants.

	Full Model *N* = 212		Adjusted Full Model *N* = 212		Sensitivity Analysis 1 ^#^ *N* = 197		Sensitivity Analysis 2 ^‡^ *N* = 34		
Variables	β (SE)	*p*-Value	β (SE)	*p*-Value	β (SE)	*p*-Value		β (SE)	*p*-Value
Time		**0.002**		**0.02**		**0.005**			0.12
Midpoint	−0.47 (0.20)		−0.44 (0.24)		−0.52 (0.23)			−0.55 (0.46)	
Endpoint	−0.74 (0.21)		−0.66 (0.23)		−0.76 (0.24)			−0.94 (0.45)	
DUFB use	1.13 (0.35)	**0.001**		**0.02**	1.38 (0.38)	**<0.001**	Frequent use	0.74 (0.49)	0.14
Time DUFB use		0.43		0.53		0.54	Time∗Frequent use		0.94
Baseline Yes	1.28 (0.41)		1.28 (0.51)		1.53 (0.45)			0.89 (0.81)	
Midpoint Yes	1.32 (0.42)		1.02 (0.53)		1.56 (0.46)			0.76 (0.72)	
Endpoint Yes	0.81 (0.44)		0.76 (0.51)		1.06 (0.49)			0.58 (0.62)	

DUFB Double Up Food Bucks; Adjusted model controlled for age, gender, race/ethnicity, household size, income, and education ^#^ This model was run without the 15 participants who had previously used the DUFB program before the study, *N* = 197. **^‡^** This model compares DUFB users who used the program more frequently (*N*=10) to those who used it less frequently (*N* = 24). Bolded values highlight *p*-values ≤ 0.05.

## Data Availability

Data supporting the reported results can be provided on request by contact with the corresponding author.

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
