# Peer review of "Higher Fruit and Vegetable Intake Is Associated with Participation in the Double Up Food Bucks (DUFB) Program"

_nutrients, 2021, doi:10.3390/nu13082607_

Round 1
Reviewer 1 Report
The authors have adequately revised the manuscript. I do have tow additional editing suggestions:
Lines 89-91: The new sentence added uses the terms participation, program, and participants. I think the first use of participation and program refer to DUFB and the use of participants refers to the study participants. Would the authors please clarify?
Figure 1: Add a legend to identify which line represents users vs nonusers.
Reviewer 2 Report
I have reviewed the paper. I am satisfied with the revised manuscript.
Author Response
Thank you for your time and review.
This manuscript is a resubmission of an earlier submission. The following is a list of the peer review reports and author responses from that submission.
Round 1
Reviewer 1 Report
This study offers to contribute to the understanding of the role that incentive programs like the Double Up Food Bucks (DUFB) program play in influencing fruit and vegetable intake among SNAP users. The study has many merits, but clarification of some key aspects is highly recommended.
- Intervention: Please clarify what constitutes the intervention. Based on the description in the methods section, it appears that the DUFB program is the intervention. However, in the discussion, the "informational strategy" aspect of the intervention is highlighted as critical to the intervention's success. This is not described in the methods section.
- Study design: Please clarify how the study groups were formed - was it simply a matter of grouping those who did and did not participate in the DUFB program? The authors state that this is a quasi experimental study, but even so, participants were not "assigned" to the intervention or control group. Was the control group exposed to aspects of the intervention (e.g., informational strategy)? If so, the authors might consider referring to the study groups as users vs nonusers. Control would be misleading in this case.
Additional comments:
- Intervention: Please describe in greater detail how the DUFB program functions - what determines the dollar equivalent that the participant is eligible to receive? Does this potentially influence how impactful the DUFB program may be on F&V intake?
- Measures - was household size assessed? This is of interest given how food purchasing decisions and consumption patterns are influenced by the household composition. For example, beyond the study participant, did children or other members of the household change their F&V intake?
Author Response
|
Reviewer Comment |
Response |
|
This study offers to contribute to the understanding of the role that incentive programs like the Double Up Food Bucks (DUFB) program play in influencing fruit and vegetable intake among SNAP users. The study has many merits, but clarification of some key aspects is highly recommended.
|
Thank you for your review and suggested revisions, they have made the paper stronger. |
|
· Intervention: Please clarify what constitutes the intervention. Based on the description in the methods section, it appears that the DUFB program is the intervention. However, in the discussion, the "informational strategy" aspect of the intervention is highlighted as critical to the intervention's success. This is not described in the methods section. |
· Thank you for your comments, the primary intervention of interest is the DUFB program. However, because previous work indicated that a main barrier to participation was unawareness, we included an informational strategy during the baseline data collection. We have added a sentence to reflect awareness created among the study participants at baseline, see lines 82-86.
|
|
· Study design: Please clarify how the study groups were formed - was it simply a matter of grouping those who did and did not participate in the DUFB program? The authors state that this is a quasi experimental study, but even so, participants were not "assigned" to the intervention or control group. Was the control group exposed to aspects of the intervention (e.g., informational strategy)? If so, the authors might consider referring to the study groups as users vs nonusers. Control would be misleading in this case. |
Thank you for your comment. Upon further reflection, we believe the term natural experiment better describes our study. The group assignment was not done by the researchers, rather the participants chose whether or not to participate, and we observed. We have replaced quasi-experimental throughout. I believe we can consider the non-users a control group for the DUFB intervention, but not for the informational strategy intervention. To avoid confusion, we have used your strategy of calling the groups users and non-users throughout.
|
|
Intervention: Please describe in greater detail how the DUFB program functions - what determines the dollar equivalent that the participant is eligible to receive? Does this potentially influence how impactful the DUFB program may be on F&V intake? |
Please see lines 75-80. All SNAP recipients were eligible for up to $10 per visit. The amount was matched dollar-for-dollar to the amount of SNAP spent. We have revised these sentences to try to improve clarity. It is true that a larger incentive, an incentive scaled for family size, or even other models may have a greater impact on fruit and vegetable consumption, but exploring those questions is outside of the scope of this study. |
|
· Measures - was household size assessed? This is of interest given how food purchasing decisions and consumption patterns are influenced by the household composition. For example, beyond the study participant, did children or other members of the household change their F&V intake? |
Yes, we agree this is true. We have added household size to our demographics table, and have run an additional model controlling for household size and other demographics which you can now find in table 3.
Unfortunately, it was beyond the scope of this study to measure intake of other household members. |
Reviewer 2 Report
This paper is well constructed, the introduction and methods are clearly written.
And as it stands, it is well laid out.
However, there are several issues that can be addressed to improve the paper.
Line 11: fruit and vegetable (F&V)
Line 13: F&V
Lines 51-53: Citation?
Line 56: Delete “finally”
Line 76: FM
Line 79: EBT: elaborate it..
Line 81: F&V
Line 86: Revise the sentence
Line 87: An a?
Lines 87-88: Reference? Or show the calculation here.
Line 93: 410 participants--- consideration of % of subject increment to avoid follow-up loss?
Line 102: 111 participants: was the number adequate to determine differences with optimal power?
Line 167: Correction of the significance level for multiple comparisons?
Figure 1: line indicator?
Line 265: time.
Line 290: a self-reported
Line 377: participants
Justification is needed or mention in limitation to consider the significant results cautiously because of having imbalance intervention and control groups (34 vs 178).
Consideration of demographics variables as covariates (gender, ethnicity, education, household income) in the model?
Thank you.
Author Response
|
Reviewer Comment |
Response |
|
This paper is well constructed, the introduction and methods are clearly written. However, there are several issues that can be addressed to improve the paper. |
Thank you for your kind comments and suggested changes, they have improved the paper. |
|
Line 11: fruit and vegetable (F&V)
|
F&V was added
|
|
Line 13: F&V |
F&V substituted |
|
Lines 51-53: Citation? |
Citation was added |
|
Line 56: Delete “finally” |
Finally was deleted |
|
Line 76: FM |
FM was added |
|
Line 79: EBT: elaborate it.. |
EBT was spelt out |
|
Line 81: F&V |
F&V substituted |
|
Line 86: Revise the sentence : |
We are unclear what the concern with this sentence was. We revised the sentence to improve clarity. |
|
Line 87: An a? |
It is “An a priori power calculation . . .” we’ve added italics to improve clarity. |
|
Lines 87-88: Reference? Or show the calculation here. |
Thank you for your comment. We’ve added a reference, current line 91. |
|
Line 93: 410 participants--- consideration of % of subject increment to avoid follow-up loss? |
It is unclear what the reviewer means here. Please see earlier in the paragraph that we have assumed a 40% retention rate, and lines 358-363 for a discussion of efforts to avoid loss of participants to follow-up. |
|
Line 102: 111 participants: was the number adequate to determine differences with optimal power? |
An after the fact power calculation is not recommended by our statistician. However, we do discuss the fact that the targeted sample size was not reached on lines 346-348. An additional sentence to make this limitation clear is added in line 348. |
|
Line 167: Correction of the significance level for multiple comparisons? |
We do not think it is appropriate to adjust the significance level for the comparisons in table 1 and 2, because for these comparisons a false negative (the groups are truly different but our text indicates no significant difference) is much more concerning than a false positive. For the multi-level regression model, the models are automatically adjusted for multiple comparisons. Although we do present four different models, we don’t believe that it is necessary to adjust for multiple comparisons, since the results were consistent between the full, adjusted, and sensitivity analysis 1. Although sensitivity analysis 2 shows a different pattern of results, we do not have any positive findings, so the chances of a false-positive due to multiple comparisons is low. |
|
Figure 1: line indicator? |
Thank you, the legend has been added. |
|
Line 265: time. |
Period added, thank you |
|
Line 290: a self-reported |
“an self-reported” was changes to “a self-reported” |
|
Line 377: participants. |
We are referring to the decision of an individual to use or not to use SNAP here. |
|
Justification is needed or mention in limitation to consider the significant results cautiously because of having imbalance intervention and control groups (34 vs 178). |
Thank you for your comment, we added a sentence on that in line 341. |
|
Consideration of demographics variables as covariates (gender, ethnicity, education, household income) in the model? |
Thank you for this suggestion, we have run an additional model to examine this question. We have included the results of an adjusted model in our table 3, the results section, and the analysis details are now found in the method section, lines 152-153. |